# Discrete Diffusion for Co-Speech Gesture Synthesis

Ankur Chemburkar
Shuhong Lu
Andrew Feng
chemburk@usc.edu
shuhongl@usc.edu
feng@ict.usc.edu
Institute for Creative Technologies, University of Southern California
Los Angeles, California, United States

## ABSTRACT

In this paper, we describe the gesture synthesis system we developed for our entry to the GENEA Challenge 2023. One challenge in learning the co-speech gesture model is that there may be multiple viable gesture motions for the same speech utterance. Therefore compared to a deterministic regression model, a probabilistic model will be preferred to handle the one-to-many mapping problem. Our system utilizes the vector-quantized variational autoencoder (VQ-VAE) and discrete diffusion as the framework for predicting co-speech gestures. Since the gesture motions are produced via sampling the discrete gesture tokens using the discrete diffusion process, the method is able to produce diverse gestures given the same speech input. Based on the user evaluation results, we further discuss about the strength and limitations of our system, and provide the lessons learned when developing and tuning the system. The subjective evaluation results show that our method ranks in the middle for human-likeness among all submitted entries. In the the speech appropriateness evaluations, our method has preferences of 55.4% for matched agent gesture and 51.1% for matched interlocutor gestures. Overall, we demonstrated the potential of discrete diffusion models in gesture generation.

## CCS CONCEPTS

• **Computing methodologies → Intelligent agents**; **Animation**; *Neural networks.*

## KEYWORDS

gesture synthesis, computer animation, neural networks

**ACM Reference Format:**
Ankur Chemburkar, Shuhong Lu, and Andrew Feng. 2023. Discrete Diffusion for Co-Speech Gesture Synthesis. In *INTERNATIONAL CONFERENCE ON MULTIMODAL INTERACTION (ICMI '23 Companion), October 9–13, 2023, Paris, France.* ACM, New York, NY, USA, 7 pages. https://doi.org/10.1145/3610661.3616556

## 1 INTRODUCTION

Co-speech gesture synthesis is an important capability for driving virtual character movements in conversational interactions with human users. It plays an essential role in augmenting the virtual human with non-verbal behaviors that mimic actual human communications in addition to speech lip-syncing animations. However, it is not trivial to synthesize gesture motions that are both human-like and correspond well to the speech input.

In general, the process of gesture generation from speech to motion is a non-deterministic one-to-many mapping, which indicates that multiple gestures could correspond to the same speech input to convey a similar meaning. For example, a left-hand beat, a right-hand beat, or a beat involving hands will all be appropriate representations of a beat motion corresponding to an utterance. Therefore instead of using deterministic models [13, 40, 41] to predict gestures, the recent methods utilized the probablistic frameworks [2, 23] by sampling the latent space to accommodate the non-deterministic natures of gesture synthesis.

For the GENEA challenge [21], we have developed our gesture synthesis system based on vector-quantized variational autoencoder (VQ-VAE) and denoising diffusion probabilistic models. We assume that by utilizing the discrete tokens, the gesture synthesis problem could be regarded as token sampling based on the predicted logits. This allows gestures that are far apart in the motion space to be still mapped to the same input utterance. By leveraging the disentanglement of information in the latent space of VQ-VAE, the system gains the potential for controllable gesture synthesis. The diffusion methods have been adapted successfully for various applications including image and motion synthesis [10, 35, 44]. The motivation for our system is to utilize these recent developments in generative models for gesture synthesis. One more insight for employing the diffusion process is that diffusion models are inherently robust to noise and uncertainty in the data. We aim to reduce jittering results generated by many previous methods. Diffusion can effectively denoise corrupted inputs by stepping backward through the diffusion process, aiding in data recovery and reconstruction tasks. Specifically, we first learn the discrete latent codes from the input motions using VQ-VAE. These codes are then used by the discrete denoising diffusion probabilistic models (D3PM) to learn the denoise process. By learning the denoising model in the discrete latent space, the method is able to leverage the synthesis strength from the diffusion process while also greatly reducing the computational costs by requiring much fewer diffusion steps to converge. After predicting the discrete codes, the model then reconstructs

the gesture motions through the decoder of VQ-VAE. From the synthesis results, we found that the method is able to produce diverse gestures with good motion dynamics. A demonstration video showcasing our results can be accessed by visiting the provided link: here."

## 2 BACKGROUND

### 2.1 Co-Speech Gesture Synthesis

In the realm of speech gesture synthesis, traditional rule-based approaches have relied on manually created sets of gesture units, employing predefined rules and heuristics to generate gestures based on linguistic and contextual information [5, 19, 25]. Some approaches have attempted to extract gesture units from training speech-gesture pairs [12, 16]. However, these methods have struggled in accurately estimating gesture attributes and effectively forming units, thereby impacting the final quality of results.

In contrast, learning-based approaches have emerged, wherein certain methods utilize speech-gesture pair data to train end-to-end models that directly predict co-speech gestures, treating the task as a regression problem from speech to gestures [6, 14, 20, 40]. However, a significant challenge arises when a single speech input corresponds to multiple variants of gestures, as the regression model tends to average the gesture poses, resulting in inferior outcomes. This challenge is commonly referred to as the one-to-many mapping from speech to gestures issue.

Recent advancements have approached gesture synthesis in a probabilistic framework, enabling the generation of multiple gesture sequences from a single speech input through latent space sampling [1, 2, 7, 23, 24, 27]. Nonetheless, as the length of the sequence increases, the process of generating data sequentially becomes time-consuming, and the dependency information is lost as each element relies on the previously generated ones [29].

Based on the aforementioned points, we propose our model that combines the VQ-VAE and diffusion techniques to tackle these challenges and enhance the synthesis of speech gestures.

### 2.2 Discrete Latent Space Learning

A VAE (Variational Autoencoder) is a type of generative model that learns a compressed representation of input data by mapping it to a lower-dimensional latent space, typically modeled as a Gaussian distribution, using an encoder. In the case of VQ-VAE, the latent space is discretized into a finite set of codebooks [36]. This allows for the encoding of original gestures into small, trainable data units using vector quantization. Recent model design and training techniques have been focusing on improvements for learning the latent space reconstructions. For instance, Jukebox [9] trained separate VQ-VAEs on data with different resolutions by hierarchically downsampling the input data. RQ-VAE [30] reduces the reconstruction errors by recursively quantizing the feature maps using a fixed-size codebook.

One known issue in VQ-VAE is codebook collapse [30], where multiple embeddings in the codebook collapse and become identical or nearly identical during training. This collapse leads to a loss of diversity in learned representations and can adversely affect model performance and generation quality. Several techniques have been proposed to mitigate codebook collapse, including re-initializing

unused codes to random vectors during each training iteration [9], normalizing mean squared error (MSE) for reconstruction [39], and updating codebook embeddings with exponential moving averages [30].

VQ-VAE method typically utilizes autoregressive transformers to learn a probability distribution over the latent space during the generative stage. However, autoregressive models often struggle with capturing long-range dependencies in the data, as each element's conditioning is limited to the previous elements. In this work, we instead applied discrete diffusion to enlarge the sampling window size without negatively affecting the performance of the generated sequences.

### 2.3 Denoising Diffusion Probabilistic Models

Diffusion models have emerged as a prominent approach in image synthesis and motion generation, showcasing their ability to generate complex and realistic results. In contrast to autoregressive generative models, diffusion models provide greater flexibility with reduced error accumulation during inference and are well-suited for parallel training since they are not constrained by step-by-step sampling [10, 17, 31–33].

In the continuous diffusion process, the target data array, such as gesture motions in our case, undergoes an iterative injection of Gaussian noise through a forward Markov process until pure noise is obtained. In the subsequent reverse process, the model learns to gradually denoise the sample. The diffusion transformer framework has found application in motion synthesis domains, including tasks like audio-conditioned gesture generation [43] that can effectively handle long-term dependencies in gesture sequences. Several notable adaptations of diffusion models have been made for human motion synthesis as well, such as generating raw motion frames [35] and improving jittering problems through time-varying weight schedules for noise estimation [8]. In the realm of gesture synthesis, Ao et al. [3] leverage a latent diffusion model and apply a Contrastive-Language-Image-Pretraining strategy [28] to learn the relationship between speech transcripts and gestures. Additionally, Zhu et al. [46] focus on ensuring temporal coherence by tailoring their Diffusion Co-Speech Gesture framework in the context of gesture synthesis.

Diffusion models can also be extended to discrete data, including categorical labels or text. For example, D3PM [4] utilizes a transition matrix in the noising step to handle discrete data. Another variant, the VQ-Diffusion model [15], combines a VQ-VAE with a conditional DDPM variant to model the latent space for text-to-image synthesis. In our system, we adapted the discrete diffusion model to produce gesture token sequences based on input conditions.

## 3 DATA PRE-PROCESSING

The training data for the GENEA Challenge 2023 is based on a subset of the Talking with Hands (TWH) dataset [22]. The dataset includes the entirety of dyadic interactions, with audio and speech text features from both the main agent and interlocutor.

In accordance with [42], we undertook analogous data preprocessing procedures. For input gesture representation, we first downsampled the input motions to 30 fps and applied a sliding window of 64 frames with a step size of 10 frames to produce gesture samples.

Each gesture sample is converted into a tensor of size $T \times J \times D$, where $T = 64$ is the sliding window size, $J$ is the number of joints, and $D$ is the size for joint rotation representation.

We also use $D = 6$ as the representation for joint rotations based on previous research [45] to prevent singularities and reduce rotation approximation errors. The pose dimension we used is 153, which includes 6D rotation vectors for 25 joints and the root translation. For each gesture sample, our target is to predict the main agent poses, and we combine the audio features from both the main agent and interlocutor as the input conditions to our model. Following the baseline data processing scripts provided by the organizers, the audio features include Mel-frequency cepstral coefficients (MFCCs), spectrogram, and speech prosody. We concatenate all three features for both agents into the final speech audio features.

## 4 METHOD

The method implemented in our system uses a two-stage architecture to train the gesture synthesis models; the first stage involves learning discrete tokens using VQ-VAE, while the second stage makes use of the discrete diffusion process to learn conditional token distributions. Figure 1 presents a summary of our approach based on discrete diffusion.

### 4.1 Discrete Gesture Token Learning

We employ a latent space vector quantization model that has been specially trained on the realm of three-dimensional human gestures. When given a human gesture represented by a sequence of poses $\mathbf{g} \in \mathbb{R}^{L \times D_g}$, where $L$ denotes the length of the gesture sequence and $D_g$ denotes the dimensions of a single gesture frame, an encoder $\mathbf{E}$ converts these frames into gesture tokens or snippets $\mathbf{s} \in \mathbb{R}^{l \times h}$, where $l$ denotes a number significantly less than $L$ and $h$ denotes the latent dimension. Then, using a discrete quantization technique $DQ$ and a learned codebook $C$ with $K$ embedding entries $(\mathbf{c}_1, ... \mathbf{c}_K)$ of dimensions $\mathbb{R}^h$, these fragments are converted into quantized vectors $\mathbf{b} \in \mathbb{R}^{l \times h}$. $DQ$ performs a transformation on $\mathbf{s}$ by comparing $(\mathbf{s}_i)_{i=1}^t$ to all codebook entries and switches the snippet with the closest codebook index. Hence, the process $DQ$ is defined as,

$$\mathbf{k}_i = argmin_{\mathbf{c}_j \in C} ||\mathbf{s}_i - \mathbf{c}_j|| \tag{1}$$

In the reverse quantization process to determine the latent embedding for each snippet, $DQ'$ transforms the indices $\mathbf{k}$ into the relevant entries $\mathbf{b}$ from codebook $C$. In the end, a decoder $D$ reconstructs $\mathbf{b}$ to the 3D space for human gestures. The general formulation of this autoencoder technique is:

$$\widehat{\mathbf{g}} = D(DQ'(DQ(E(\mathbf{g})))) \tag{2}$$

This procedure is trained with an embedding loss to update the codebook entries and stabilize training, and a reconstruction loss between $\mathbf{g}$ and $\widehat{\mathbf{g}}$ given by:

$$\mathbf{L}_{vq} = ||\widehat{\mathbf{g}} - \mathbf{g}||_1 + ||sg[E(\mathbf{g})] - \mathbf{b}||_2^2 + \beta ||E(\mathbf{g}) - sg[\mathbf{b}]||_2^2 \tag{3}$$

sg[.] stands for the stop gradient operation in this context and $\beta$ is a weighting factor. Since the quantization process $DQ$ is not differentiable, back-propagation was made possible by using the straight-through gradient estimator [37].

In our system, the encoder and decoder layers for the VQ-VAE model are a series of convolutional layers with skipped connection, which are adapted from the recent work in image synthesis [11]. Since their original applications were 2D image synthesis, we changed the 2D convolutions layers into 1D to better fit the data dimensions for the gesture motions. We use $l = L/4$ in our experiments which gives us a sequence length $l$ of 16.

### 4.2 Diffusion for Discrete Gesture Tokens

The discrete diffusion model and its continuous equivalent share many similarities. The forward diffusion process gradually corrupts the sample through a Markov chain $q(\mathbf{k}_t | \mathbf{k}_{t-1})$, given a sequence of discrete tokens $\mathbf{k}_0 \in \mathbb{I}^l$, where the subscript denotes the diffusion step. Following the discrete diffusion process [15], we employ the forward process to create progressively noisier latent variables $\mathbf{k}_1, \ldots, \mathbf{k}_T \in \mathbb{I}^l$, where $T$ represents the total number of diffusion steps. In this discrete diffusion example, $\mathbf{k}_T$ consists of pure noise or all masked tokens.

The reverse diffusion process samples from the reverse distribution $q(\mathbf{k}_{t-1} | \mathbf{k}_t, \mathbf{k}_0)$ in an attempt to reconstruct $\mathbf{k}_0$ from $\mathbf{k}_T$. To approximate the reverse distribution, we train a transformer model as the denoising model. The transformer model produces the distribution represented by the symbol $p_\theta(\mathbf{k}_{t-1} | \mathbf{k}_t, \mathbf{y})$, where $\mathbf{y}$ denotes the condition (e.g., speech/text/interlocutor gestures or their combination).

The transitional probabilities between codebook indices are defined by fixed transition matrices $Q_t \in \mathbb{R}^{(K+1) \times (K+1)}$ at each timestep. The matrix $\mathbf{Q}$ is given by,

$$\mathbf{Q}_t = \begin{bmatrix} \alpha_t + \beta_t & \beta_t & \beta_t & \dots & 0 \\ \beta_t & \alpha_t + \beta_t & \beta_t & \dots & 0 \\ \beta_t & \beta_t & \alpha_t + \beta_t & \dots & 0 \\ \vdots & \vdots & \vdots & \ddots & \vdots \\ \gamma_t & \gamma_t & \gamma_t & \dots & 1 \end{bmatrix} \tag{4}$$

The [MASK] token is represented by the extra dimension in $K + 1$. According to $Q_t$, an index in $\mathbf{k}_t$ has a probability of $K\beta_t$ of being replaced by another index chosen randomly from the $K$ indices, with a probability $\gamma_t$ of turning into a [MASK] index, and a probability of $\alpha_t$ of staying the same index at each diffusion step.

During training, the forward diffusion process becomes efficient by utilizing the closed-form equation [15] of the cumulative transition matrix $\overline{Q_t} = Q_t \ldots Q_1$, which expresses the transition probability from $\mathbf{k}_0$ to $\mathbf{k}_t$ and the corresponding forward probability distribution $q(\mathbf{k}_t | \mathbf{k}_0)$. Throughout the reverse process, the model learns to approximate the posterior $q(\mathbf{k}_{t-1} | \mathbf{k}_t, \mathbf{k}_0)$ with $p_\theta(\mathbf{k}_{t-1} | \mathbf{k}_t, \mathbf{y})$, as mentioned earlier.

To enhance generation results, recent efforts [4, 18] utilize a reparameterization approach, approximating the distribution rather than directly modeling the posterior. The denoising model produces denoised gesture tokens given by $p_\theta(\tilde{\mathbf{k}}_0 | \mathbf{k}_t, \mathbf{y})$. By using the denoised token distribution $p_\theta(\tilde{\mathbf{k}}_0 | \mathbf{k}_t, \mathbf{y})$ and the posterior distribution $q(\mathbf{k}_{t-1} | \mathbf{k}_t, \tilde{\mathbf{k}}_0)$, we sample the $(t-1)$-th gesture from $p_\theta(\mathbf{k}_{t-1} | \mathbf{k}_t, \mathbf{y})$ during inference.

The diffusion model is implemented as a transformer architecture [38] with 19 layers and 16 attention heads. We use 100 diffusion

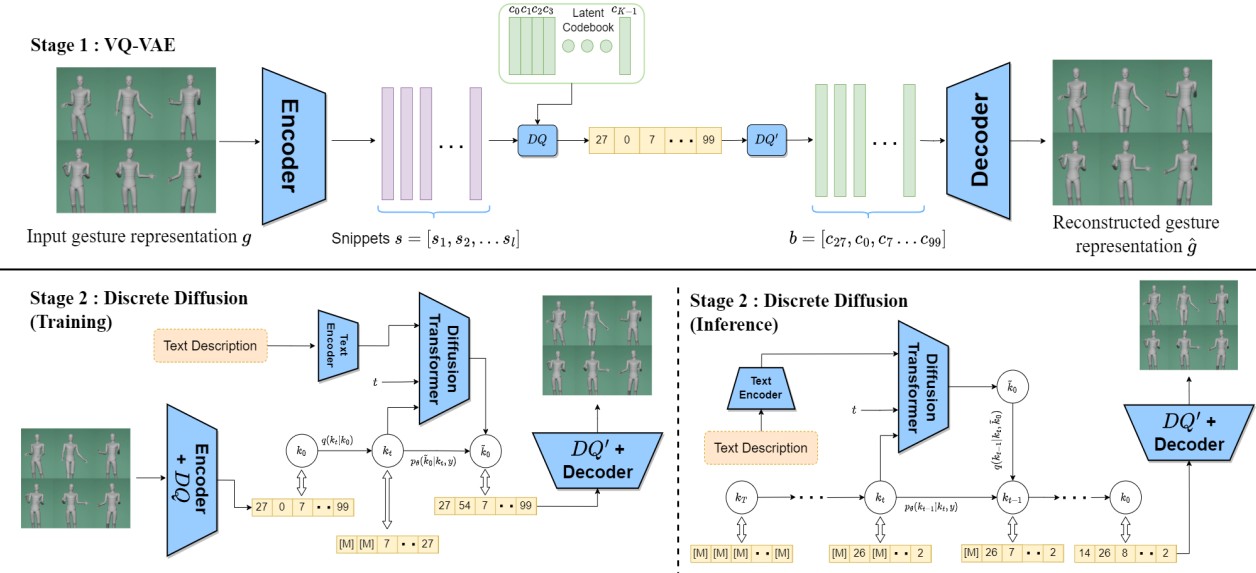

**Figure 1: Architecture for VQ-Diffusion model. The top half represents the VQ-VAE model framework. Bottom left figure briefly shows the forward and reverse process of the training stage in Diffusion. Bottom right figure explains the inference stage with the reparametrization trick.**

steps for our method and set the condition hidden dimension as 512.

## 4.3 Classifier-Free Guidance

The diffusion model attempts to optimize the prior distribution $p(\mathbf{k}|\mathbf{y})$ during the training phase of a conditional generation task using $\mathbf{k}$ as a sample and $\mathbf{y}$ as the associated condition, provided that the posterior distribution $p(\mathbf{y}|\mathbf{k})$ is satisfied. It's probable that throughout training, this posterior probability will be disregarded. It is possible that the model merely uses the corrupted sample to reconstruct and ignores the conditional input because it has access to both the corrupted sample and the condition. The *posterior issue* [34], or poor alignment between the generated sample and the condition, results from this.

Therefore, both $p(\mathbf{k}|\mathbf{y})$ and $p(\mathbf{y}|\mathbf{k})$ must be included in our optimization objective. One way to do this is to optimize $\log p(\mathbf{k}|\mathbf{y}) + s \log p(\mathbf{y}|\mathbf{k})$, where $s$ denotes the guidance scale which is a hyperparameter. By using Bayes' Theorem, this optimization function can be expressed as:

$$argmax_{\mathbf{k}} = [\log p(\mathbf{k}) + (s+1)(\log p(\mathbf{k}|\mathbf{y}) - \log p(\mathbf{k}))] \quad (5)$$

where $p(\mathbf{k})$ is the unconditional distribution of $\mathbf{k}$. To handle the unconditional inputs, the model is also trained with a 'null' condition [26] for a select percentage of samples. It has been shown that implementing a learnable conditional vector instead of a 'null' condition is more suitable for training classifier-free guidance [34]. We adopt the technique with a learnable null vector in our implementation. Empirically, we found that using the classifier-free guidance with a proper guidance scale improves the overall gesture synthesis results.

## 5 RESULTS AND DISCUSSION

### 5.1 Implementations and Experiments

We chose to train VQ-VAE over 35k steps (120 epochs) on a batch size of 256 which takes approximately 90 minutes to show proper convergence. The VQ-VAE model was trained with both the L2 reconstruction loss and the codebook loss. In addition, we utilized Fréchet Gesture Distance (FGD) as the perceptual metric to evaluate whether the reconstructed motions were statistically faithful to the original motion styles. Figure 2 (Top row) shows the loss graphs for training the VQ-VAE, which demonstrates the method is capable of learning the discrete representation and reconstructing the original gestures. The VQ-VAE model shows good gesture reconstruction capabilities as proven by the best validation FGD of 0.7. However, empirically we observed one peculiarity that using the VQ-VAE model with the best reconstruction FGD may produce worse results when training the discrete diffusion model in the 2nd stage. We suspected this may be due to overfitting and thus chose a VQ-VAE checkpoint with FGD of 1 for training the discrete diffusion model.

For training the 2nd stage diffusion model, the KL divergence loss was used since the diffusion is operated on the discrete labels. For selecting the best checkpoint, FGD was also used as the evaluation metric to reflect the motion quality of synthesized gestures. During training, the discrete diffusion model converged with a steady decrease in KL loss until the model started to overfit at around 12K steps again on a batch size of 256. The FGD was also converging smoothly without large fluctuations as shown in Figure 2 (Bottom row). As seen in the plots, FGD continued to improve despite the increase in validation loss. Therefore for stage 2, we picked the checkpoint with the lowest FGD since it was observed

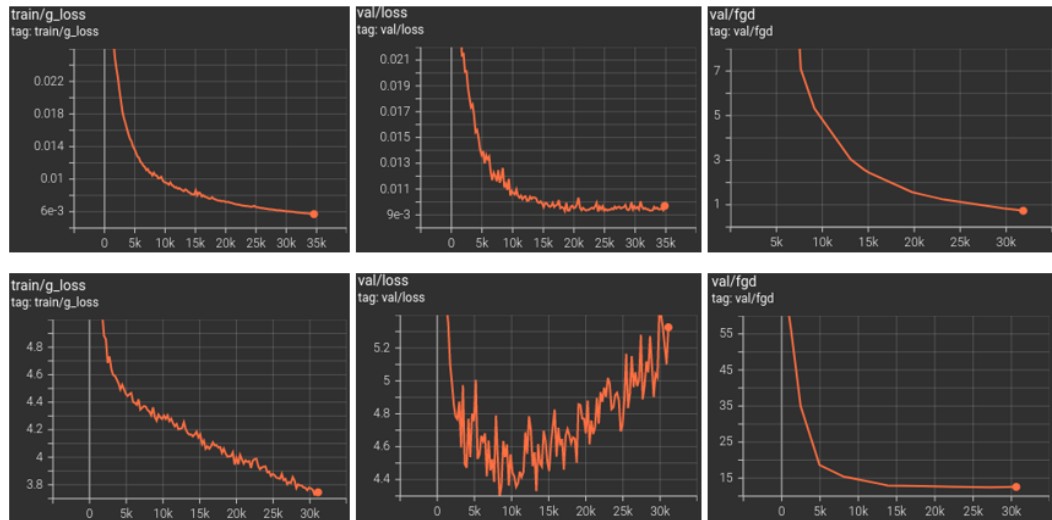

**Figure 2: Metric plots on the Genea2023 dataset training and validation. Top row shows the metrics for training and validating of the VQ-VAE stage with training loss, validation loss and FGD from left to right. Bottom row shows the metrics for diffusion model trained and validated on the above VQ-VAE. Once, again with training loss, validation loss and FGD from left to right.**

empirically that the overfitted model with lower FGD resulted in better-looking gestures.

## 5.2 Subjective Evaluations

The user study and evaluations were conducted by the GENEA 2023 organizers. The videos for the subjective evaluations were rendered from the gesture motion submissions from each team. Since the challenge dataset is based on dyadic conversations between two agents, three tasks were evaluated to properly assess different qualities for the generated gesture motions. The Human-likeness study measures the overall quality of the generated motions without factoring in the speech content. Appropriateness for agent speech study measures whether the synthesized gestures correspond well to the input speech without considering the interlocutor. Finally, appropriateness for the interlocutor includes the dyadic interactions to evaluate whether the interlocutor's motions are proper given the conversations and the main agent's motions. In the following, we further discuss the evaluation results for our system (SI).

Figures 3, 4a, 4b show the subjective evaluations of various models on the test dataset. Our model (SI) shows average performance and ranks in the middle of all competing models. The average result can be attributed to a few reasons. First, due to the efforts for developing and tuning the VQ-diffusion model, we were not able to perform extensive experiments with all different input conditions within the timeline for the Challenge. Therefore the model has been conditioned only on the audio of the main agent and interlocutor for simplicity in the experiments. The possible improvement would be including additional conditions such as the text transcript for better speech context, interlocutor gestures for more appropriate dyadic gestures and speaker identities for varying the gesture styles of different speakers. A combination of these input features can be fused with the audio features in a joint embedding space which could serve as a better conditional input for diffusion. Another

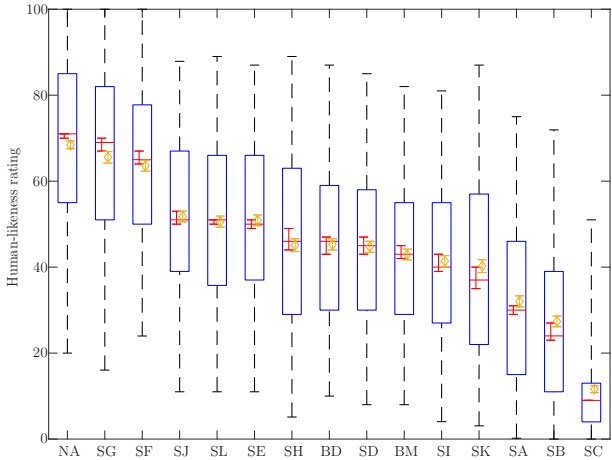

**Figure 3: Box plot visualising the ratings distribution in the human-likeness study. Red bars are the median ratings (each with a 0.05 confidence interval); yellow diamonds are mean ratings (also with a 0.05 confidence interval). Box edges are at 25 and 75 percentiles, while whiskers cover 95 % of all ratings for each condition. Conditions are ordered by descending sample median rating.**

reason for the average performance is that we have ignored synthesizing the finger joints when training our models, and focused only on producing the body and arm motions. Including these additional finger motions would likely enhance the details of the gestures and boost the overall motion quality in the subjective evaluations.

Moreover, on inspection of our generated gestures visually, we observed a jittering issue in some results. Specifically, sometimes the synthesized gesture motions may produce abrupt movements

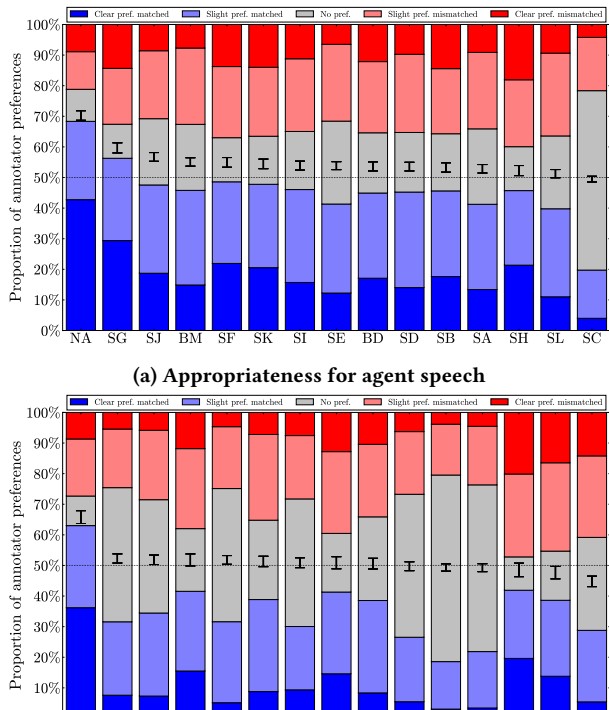

(a) Appropriateness for agent speech

(b) Appropriateness for the interlocutor

**Figure 4: Bar plots visualising the response distribution in the appropriateness studies. The blue bar (bottom) represents responses where subjects preferred the matched motion, the light grey bar (middle) represents tied ("They are equal") responses, and the red bar (top) represents responses preferring mismatched motion, with the height of each bar being proportional to the fraction of responses in each category. Lighter colours correspond to slight preference, and darker colours to clear preference. On top of each bar is also a confidence interval for the mean appropriateness score, scaled to fit the current axes. The dotted black line indicates chance-level performance. Conditions are ordered by mean appropriateness score.**

that look like noises and motion artifacts. Originally we thought this was due to the singularity of the pose representation. However, the jittering still persisted after we switched to the 6-D rotation representation. Therefore we speculated that the possible reason for this effect could be due to the discrete nature of the representation. During the learning process, the discrete diffusion process might have predicted to shift between codebook indices representing two very different gestures. Even though the VQ-VAE decoder should alleviate the discontinuous motions, this may still lead to sudden speed changes in the gesture being performed and reduces the overall smoothness of the produced motion. Resolving this issue requires a deeper investigation into the diffusion model training to understand the cause. Some heuristics could also be implemented to prevent sampling the subsequent gesture tokens that are too far away in the motion space.

While we believe the proposed architecture of discrete conditional diffusion is a promising method, a significant disadvantage to this method is having to train two different models. It requires training both the VQ-VAE model for learning the discrete latent codes and the discrete diffusion model for learning the conditional inference. Thus the performance of the diffusion model depends heavily on the quality of VQ-VAE and slight variance in VQ-VAE can lead to significant performance differences in the final performance.

In our experiment, we found that the codebook size of the VQ-VAE is also an important factor and it is easy to overfit if a large codebook size is chosen. For example, using a codebook size of 1024 produces worse results than a codebook size of 256, which was used in our final model. Another hyperparameter requires tuning in the guidance scale in the diffusion process. The final quantitative results vary significantly on the guidance scale. We found a guidance scale of 4 to give the best results.

## 6 CONCLUSIONS AND TAKEAWAYS

In this paper, we describe the gesture synthesis method of our submission entry to GENEA Challenge 2023 [21]. Overall, the discrete diffusion method is able to leverage the generative strength of the diffusion process while reducing the inference time compared to running the diffusion on the full motion poses. However, the user study results showed that there is still room for improvement in our proposed system. In the future, we plan to address the issues of jittering artifacts and finger motions to improve the overall motion quality. We also hope to experiment with additional input conditions to produce proper motions in dyadic scenarios. We believe the method requires more refinements and could be a promising direction for generating stylized gestures using various input conditions such as audio, text, and speaker identities once these drawbacks are addressed.

## 7 ACKNOWLEDGMENT

This work is supported by University Affiliated Research Center (UARC) award W911NF-14-D-0005. Statements and opinions expressed and content included do not necessarily reflect the position or the policy of the Government, and no official endorsement should be inferred.

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
