# OpenReview forum: "Discrete Diffusion for Co-Speech Gesture Synthesis"
_ACM.org/ICMI/2023/Workshop/GENEA_Challenge — GENEA Challenge 2023 Workshopproceeding_

### Official Review · Reviewer_xzZG · 2023-07-22
**Overall this paper describes the adequate details of their submitted system, including data processing, methods, and training details. However, minor revision is needed to improve the paper quality.**

**Rating:** 6
**Confidence:** 4

**Review:**


1. It is strongly advised that the key findings and results be summarized in the abstract (1~2 sentences). How does this system perform in the challenge?

2. I would appreciate the effort put into the descriptions of methods and implementation details. But I think the readers might be confused about the inputs/outputs of the system, by looking at Fig. 1. Audio processing is described in Section 3, but are the audio features used as input in the architecture?

3. The baseline method should be cited, as it is used for data processing.

4. There are some minor errors in this paper. I would suggest that authors do a round of meticulous checks for revision.
Line 28: "is __an__ non-deterministic one-to-many mapping"
Line 38: "the predicted __the__ logits."

5. I would love to see more insights from this submission entry. For example, are there any other reasons than one2many mapping why VQVAE+Diffusion is chosen? or how is the result compared with the baseline?

---

### Official Review · Reviewer_YkuE · 2023-07-30
**This paper describes a method for diverse co-speech gesture synthesis using discrete diffusion process. The authors have discussed the benefits of using discrete diffusion process for this task and gives a reasonable explanation of their design choices and experiments. Overall the work is technially sound, the method is well organized and experiments are clearly discussed.**

**Rating:** 6
**Confidence:** 3

**Review:**

 The paper is well-written, the sections are well-organized, and the contribution is clear.
 The paper is technically sound. The authors have motivated their reason for using discrete diffusion for such a task and discussed the strength and current limitations of their method.

A few points to consider:
1.  Details on the design choices for discrete diffusion should be included. Do the authors use the D3PM as it is for this task?
2. Ablation without the classifier-free guidance should be included to provide a justification of how a proper guidance scale improves the overall gesture synthesis results.
3. Video results would have been beneficial to judge the visual quality of the method.



**Nominate For A Reproducibility Award:**

No comments as the authors have not uploaded their code yet.

---

### Decision · Program_Chairs · 2023-08-04

**Decision:**

Accept (Workshop proceeding)

**Comment:**

All the reviewers recommended accepting this paper. The reviewers found the work to be technically sound, the method - well organized, and experiments - clearly discussed. Based on the reviews, the organisers accept this paper for the Workshop ICMI track to be published in the Adjunct ICMI Proceedings.
Please read the reviews carefully and revise the paper for the camera-ready version as follows:
1. summarize the key findings and results in the abstract
2. cite the baseline method, as it was used for data processing
3. cite the challenge paper from the organisers as instructed previously
4. do a round of meticulous checks for revision both in terms of grammar and references
5. discuss more insights from this submission entry
6. provide qualitative results, for example, by linking to a video.